# Establishment of chemosensitivity tests in triple-negative and BRCA-mutated breast cancer patient-derived xenograft models

Hyung Seok Park[1ʘ], Jeong Dong Lee[2ʘ], Jee Ye Kim[1], Seho Park[1], Joo Heung Kim[1], Hyun Ju Han[3], Yeon A. Choi[3], Ae Ran Choi[3], Joo Hyuk Sohn[4]*, Seung Il Kim[1]*

**1** Department of Surgery, Yonsei University College of Medicine, Seoul, Korea, **2** Department of Human Biology and Genomics, Brain Korea 21 PLUS Project for Medical Sciences, Yonsei University College of Medicine, Seoul, Korea, **3** Avison Biomedical Research Center, Yonsei University College of Medicine, Seoul, Korea, **4** Division of Medical Oncology, Department of Internal Medicine, Yonsei University College of Medicine, Seoul, Korea

ʘ These authors contributed equally to this work.
* SKIM@yuhs.ac (SIK); ONCOSOHN@yuhs.ac (JHS)

**Data Availability Statement:** All relevant data are within the paper.

**Funding:** This work was supported by a Faculty Research Grant of Yonsei University College of

## Abstract

### Purpose

A patient-derived xenograft (PDX) model is an in vivo animal model which provides biological and genomic profiles similar to a primary tumor. The characterization of factors that influence the establishment of PDX is crucial. Furthermore, PDX models can provide a platform for chemosensitivity tests to evaluate the effectiveness of a target agent before applying it in clinical trials.

### Methods

We implanted 83 cases of breast cancer into NOD.Cg-Prkdcscid Il2rgtm1Sug/Jic mice, to develop PDX models. Clinicopathological factors of primary tumors were reviewed to identify the factors affecting engraftment success rates. After the establishment of PDX models, we performed olaparib and carboplatin chemosensitivity tests. We used PDX models from triple-negative breast cancer (TNBC) with neoadjuvant chemotherapy and/or germline BRCA1 mutations in chemosensitivity tests.

### Results

The univariate analyses (p<0.05) showed factors which were significantly associated with successful engraftment of PDX models include poor histologic grade, presence of BRCA mutation, aggressive diseases, and death. Factors which were independently associated with successful engraftment of PDX models on multivariate analyses include poor histologic grade and aggressive diseases status. In chemosensitivity tests, a PDX model with the BRCA1 L1780P mutation showed partial response to olaparib and complete response to carboplatin.

Medicine grant no. 6-2017-0072 to HSP, a
Severance Surgeon's Alumni Research Grant grant
no. 2016-01 to HSP, National Research Foundation
of Korea grant 2018R1A2A2A15019814 to SIK,
Korea Health Industry Development Institute grant
HI14C1324 to JHS, National Research Foundation
of Korea grant 2016R1D1A1B03934564 to HSP,
and a Faculty Research Grant of Yonsei University
College of Medicine grant no. 6-2010-0002 to SIK.
The funders had no role in study design, data
collection and analysis, decision to publish, or
preparation of the manuscript.

**Competing interests:** HSP has received honoraria
from Aastrazenca, Dakeda, Ethicon, and Intuitive
Surgical. JDL, JYK, SP, JHK, HJH, YAC, ARC, JHS,
and SIK have nothing to declare. The authors
would like to declare the following patents/patent
applications associated with this research:
Germline Pathogenic Mutation of BRCA1, L1780P
(Patent pending, reference number; DPB172272).
This does not alter our adherence to PLOS ONE
policies on sharing data and materials.

## Conclusions

Successful engraftment of PDX models was significantly associated with aggressive diseases. Patients who have aggressive diseases status, large tumors, and poor histologic grade are ideal candidates for developing successful PDX models. Chemosensitivity tests using the PDX models provide additional information about alternative treatment strategies for residual TNBC after neoadjuvant chemotherapy.

## Introduction

Drug development is facilitated by understanding of the interactions between tumors and microenvironments. Preclinical models enhance knowledge of tumor biology and drug development to elucidate drug resistant mechanisms of tumors. Appropriate selection of preclinical models is pivotal to bridge the translational gap between drug development and clinical application.

Triple negative breast cancer (TNBC) tumors lack expression of estrogen receptor (ER), progesterone receptor (PR), and human epithelial growth factor receptor 2 (HER2). Compared to hormone receptor-positive breast cancer, TNBC exhibits aggressive clinical characteristics [1]. Cytotoxic chemotherapy is the only systemic treatment option to manage TNBC, because of a lack of effective target therapy [2].

Pharmaceutical companies have invested in research and development (R&D) to find an effective target therapy. However, many hurdles block the development of new drugs, including the significant cost in terms of money and time. In 2000, the R&D cost per new drug, including the pre-clinical and clinical cost was, on average US 802 million dollars [3]. The mean duration of the clinical and approval phase, from investigational new drug application filing to new drug application submission and approval, was 87.4 months [4]. Although new agents have been discovered, the successful clinical trial process is difficult [5]. A new strategy for successful clinical trial is necessary to overcome these challenges [6].

Drug sensitivity tests can be performed in in vitro models [7] whereby cancer cells are directly exposed to various drug combinations in multiple concentrations. This kind of study is considered a basic experiment before further evaluations of the drug can be tested in clinical trials [8]. However, even well-designed artificial conditions for cancer cell lines do not directly reflect patients' micro-environments [9, 10].

Mouse models, or in vivo models, use patient-derived cancer cells to induce tumors in mice. Animal models can simulate interaction between tumors and the microenvironment [11, 12]. However, in vivo models which use cultured cancer cells of in vitro studies have the same limitation because they also do not accurately reflect the characteristics of a patient's primary tumor [13–15].

Patient-derived xenograft (PDX) models which use immune-compromised mice implanted with fresh tumor tissue from patients with cancer have been introduced as new in vivo models for cancer research [16, 17]. In general, tumors in PDX models have shown similar histopathological features as primary tumors [18]. Therefore, theoretically, PDX models can predict drug response of primary tumors better than conventional in vivo or in vitro models [19–21].

Patient-derived xenograft models have been established for several cancer types [22–25] and successful take rates of PDX models vary according to cancer type. Previous studies have reported 10–50% success rates of PDX models in colon, pancreas, lung, prostate, and breast cancer [26]. To enhance successful engraftment and maintenance of PDX models, it is crucial

to characterize the factors influencing the establishment of PDX models, but little is known about those factors.

The current study focused on the establishment of PDX models of breast cancer and identifying the influencing factors for successful engraftment of the models. Furthermore, in vivo chemosensitivity tests were performed in the PDX models with TNBC, with and without highly aggressive diseases characteristics, and BRCAness.

## Materials and methods

### Patients and subtypes

A total of 82 patients with breast cancer were enrolled in this study. One patient who received serial biopsies including a preoperative needle biopsy and surgical resection before (BR30) and after (BR40) neoadjuvant chemotherapy was also included. Thus, a total of 83 malignant tumors of breast were used to develop PDX models in the study. All patients underwent breast surgery or biopsy at Severance Hospital, Seoul, Korea. Fresh tumor tissue was obtained during surgery or biopsy. After removal, tumor tissue was immediately transferred to the animal laboratory (Avison Biomedical Research Center, Seoul, Korea). Cubic tissue of $3mm^3$ in size was minced and directly implanted into mice without delay. We attempted to establish PDX models from TNBC (n = 65), luminal A or B (n = 13), and HER2 positive (n = 5) breast cancer patients. Among TNBC patients, 39 received neoadjuvant chemotherapy.

### PDX models and establishment periods

Tumor tissue was implanted into the subcutaneous or mammary fat pad of 6 to 12-week-old female NOG (NOD.Cg-Prkdcscid Il2rgtm1Sug) mice [27]. Tumor size was measured using calipers once a week, and tumor volume was calculated as $0.5 \times Length \times Width^2$[28]. When the tumor volume reached 1500 $mm^3$-, the implanted tumor was harvested from the mice under isoflurane anesthesia [26]. A portion of the tumor (~$5mm^3$) was engrafted in a new generation of mice [29]. The remaining tumor was cut into pieces and stored in liquid nitrogen. The tumor tissue was stored in a solution of 90% fetal bovine serum (FBS) and 10% dimethyl sulfoxide (DMSO) for use in re-engraftment. Some of the tumor tissues were stored frozen for generating sequencing data [30]. Primary tumor was implanted into the first generation of mice, which was defined as F1 mice. The harvested tumor was serially implanted into second- (F2) and third-generation (F3) of mice. When the PDX model had been engrafted into third generation of mice, it was considered a successful PDX model [31].

Establishment periods from F1 to F3 (F1-F3) were measured. The establishment period was divided into establishment periods from F1 to F2 (F1-F2) and F2 to F3 (F2-F3), and compared. Establishment periods from F1 to F3 (F1-F3) was not exceeded 550 days.

### Animal care

All animal experiments were performed in a facility accredited by AAALAC International (#001071) in accordance with Guide for the Care and Use of Laboratory Animals 8th edition, NRC (2010). The tumor implanted mice were housed in one cage (maximum five mice). Food was exchanged once a week and water was supplied automatically from the automatic water supply equipment. A facility staff of animal laboratory periodically checked the status of mice. Mice monitoring was performed once a week. After tumor implantation, euthanasia was performed when tumors reached $1500mm^3$ in volume or became ulcerated, tumors affected physiological functions including gait, posture, ability to eat/drink/urinate/defecate, or when mice lost more than 15% of their first body weight. Mice were euthanized by CO2 gas inhalation in

accordance with the humane endpoints which included the above descriptions. And when the mice reached endpoint criteria, we performed euthanasia by CO2 gas inhalation within 2 days. In our study, there were no mice which died before meeting criteria for euthanasia. There are well-trained facility staffs who can help to solve the problem in animal care at our institution. They advised injection methods of drugs (carboplatin/olaparib).

### Histopathological confirmation

When a tumor was engrafted to a new mouse, one piece of tumor tissue (~8mm$^3$) was used to construct FFPE (Formalin fixed paraffin embedded) blocks. Immunohistochemistry analysis was performed to confirm the histopathological similarity including ER, PR, and HER2 between primary tumor and those in PDX models. Xylene and serial concentrations of ethanol were used for deparaffinization and hydration of tumor sections. For antigen retrieval and blocking endogenous peroxidases, a solution of 0.4% $H_2O_2$ and 10mM sodium citrate buffer (pH6.0) was used. Sections were treated with a peroxidase-conjugated polymer solution (Dako REAL EnVision Detection System, Peroxides/DAB+, antibodies of Rabbit/Mouse). Primary antibodies used for in immunohistochemistry analysis included ERα (clone SP1, 1:100 dilution, LifeSpan BioSciences, Seattle, WA, USA); PR (clone 1E2, 1:100 dilution, ABCAM, Cambridge, CB4 0FL, UK), HER2 (clone EP1045Y, 1:100 dilution, ABCAM, Cambridge, CB4 0FL, UK) and Ki67 (clone EPR3610, 1:500 dilution, ABCAM, Cambridge, CB4 0FL, UK). A pathologist confirmed all stained slides.

### Factors relating to success of PDX models

Clinicopathological factors relating to the successful engraftment of PDX models were analyzed. These factors included age, T-stage, nodal status, hormone receptor status, HER2 status, Ki67 level, breast cancer subtype, histologic grade, treatment status, biopsy method (surgery or needle biopsy), engraftment period, mouse type, TNM Classification of Malignant Tumours (TNM) stage, existence/absence of *BRCA* mutation, and survival status of patient. Disease burden was analyzed. Aggressive diseases were defined as progressive diseases during preoperative or neoadjuvant chemotherapy, recurrent, and metastatic diseases.

### Chemosensitivity tests using the PDX models

We performed chemosensitivity tests using three PDX models including a tumor from TNBC with a novel germline pathogenic *BRCA1* mutation, c.5339T->C; p.Leu1780Pro; rs80357474 (L1780P), which was introduced in our previous study [32], a tumor from TNBC with a germline *BRCA1* deleterious mutation, c.3277delG; p.Val1093SerfsTer16; rs113900085, and a tumor from a TNBC without germline pathogenic *BRCA* mutation. PDX models were derived from patients who had BRCA1 mutation (n = 2, L1780P and rs113900085) and the wild type of BRCA1 (n = 1). Each model comprised nine mice, which were divided equally into three groups (1 group = 3 mouse, vehicle [PBS], olaparib, and carboplatin.). A total of 27 mice were used in the chemosensitivity tests.

Dosages of olarparib and carboplatin were 50 mg/kg and 25 mg/kg, respectively [33, 34]. Olaparib was administrated via intraperitoneal injection for 28 consecutive days. Carboplatin was administrated via intraperitoneal injection once a week. Duration of the *in vivo* test was not exceeded 2 months. Tumor volumes were measured using the below equation,

Volume = $0.5 \times$ Length $\times$ Width$^2$

## Sequencing analysis of the PDX model with L1780P

Whole genome sequencing (WGS) and whole exome sequencing (WES) of one PDX model with L1780P mutation were performed in collaboration with the Theragen company (Seoul, Korea). The depth of WGS was 30X in buffy coat, 60X in primary tumor, and 30X in F1, F2, and F3, respectively. WES was only performed in the primary tumor, and the depth of WES was 250X.

Using the TruSeq Nano DNA Sample Preparation Kit from Illumina (San Diego, CA), DNA sequencing libraries of WGS were constructed, according to the manufacturer protocol. Quality of the amplified libraries was confirmed by electrophoresis on Agilent Bioanalyzer High Sensitivity DNA Kit (part # 5067–4626) (Agilent, CA). The libraries were sequenced using Illumina HiSeq2500 and Cluster generation. Then, $2 \times 100$ cycle sequencing reads, separated by paired-end turnaround, were performed on the instrument using HiSeq Rapid SBS Kit v2 (FC-402-4021) and HiSeq Rapid PE Cluster Kit v2 (PE-402-4002; Illumina, CA).

In the WES, the quality and quantity of purified DNA were assessed by fluorometry (Qubit, Invitrogen) and gel electrophoresis, and the sample was hybridized with RNA probes, SureSelect XT Human All Exon V5 Capture library. The captured targets were then pulled down by biotinylated probe/target hybrids using streptavidin-coated magnetic beads (Dynabeads My One Streptavidine T1; Life Technologies Ltd.). The resulting purified libraries were applied to an Illumina flow cell for cluster generation and sequenced using 100 bp paired-end reads on an Illumina Hiseq2500 sequencer, following the manufacturer's protocols.

The quality of reads in the WGS and WES were confirmed using fastQC (v.0.10.1) [35], which also expounded the basic quality for sequence quality score, GC content, N content, length of distribution, and duplication levels. After examining the read quality, the low-quality bases below Q20 were trimmed using Cutadapt (v.1.8.1) [36].

In order to remove mouse reads in PDX samples, BBMap [37] was applied to the fastq files based on hg19 and Ensembl Release 77 reference genome for human and mouse, respectively. Only reads that were classified as human reads were then analyzed.

## Statistics and ethics

The SPSS statistics program version 23 (International Business Machines Crop., Armonk, NY, USA) was used for all analyses. Categorical variables were examined using chi-square test or Fisher's exact test. Continuous variables were examined using student T-test. Multivariate analyses were examined using binary regression models. Multivariate analyses were adjusted for significant factors in univariate analyses. All statistical analyses were two-sided and p-values of less than 0.05 were considered statistically significant.

All tumor tissue was obtained with the patient's written consent and the informed written consent was provided by the patients. All procedures were approved by the Institutional Review Board of Yonsei University Health System (IRB No.4-2012-0705). All experiments were approved by the Institutional Animal Care and Use Committee in Yonsei University Hospital System (YUHS-IACUC) and animals were maintained in a facility accredited by AAALAC International (#001071) in accordance with Guide for the Care and Use of Laboratory Animals 8th edition, NRC (2010).

## Results

Of the 83 tumor samples, most tumor tissue (65 out of 83 samples) came from TNBC patients (Fig 1A). Only one tumor sample was subcutaneously implanted into mice. 78 tumors were implanted into mammary fat pads. Successful engraftments of PDX models were established in 19 TNBC cases. All successful engraftments of PDX models were implanted into mammary

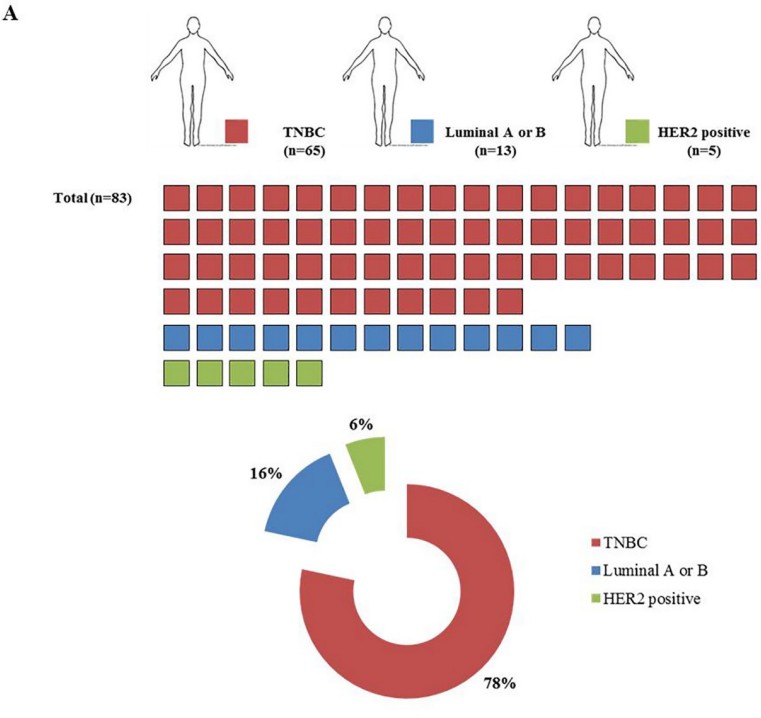

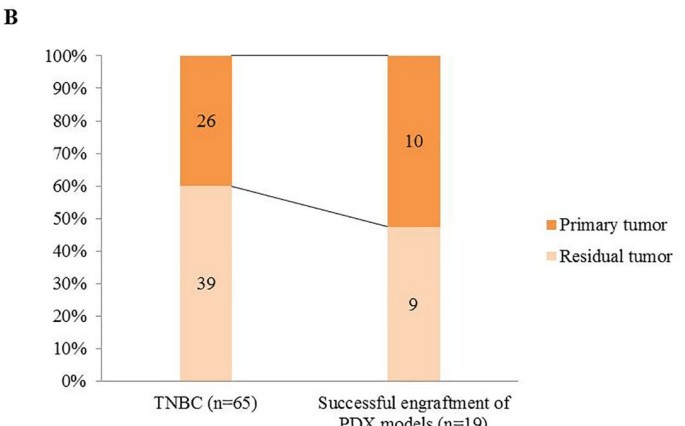

**Fig 1. Proportions of successful engraftment of PDX models.** (A) Proportion of breast cancer subtypes. (B) Successful engraftment of PDX models were derived from TNBC patient with or without neoadjuvant chemotherapy.

fat pads. We attempted to establish PDX models from residual tumors more than primary tumors (Fig 1B). The overall success rates of PDX models were higher in cases with primary TNBCs than those in residual TNBCs (38.5% for primary TNBC vs. 23.1% for residual TNBCs) (Fig 1B). We established a PDX model with a novel germline pathogenic mutation of BRCA1, L1780P (Patent pending, reference number; DPB172272).

A total of 21 successful PDX models were derived from TNBC patients. Of those, two cases were classified as developed lymphoma. Except for those two cases, 19 tumors from PDX models coincided with the histopathological characteristics of primary tumors (Fig 2A).

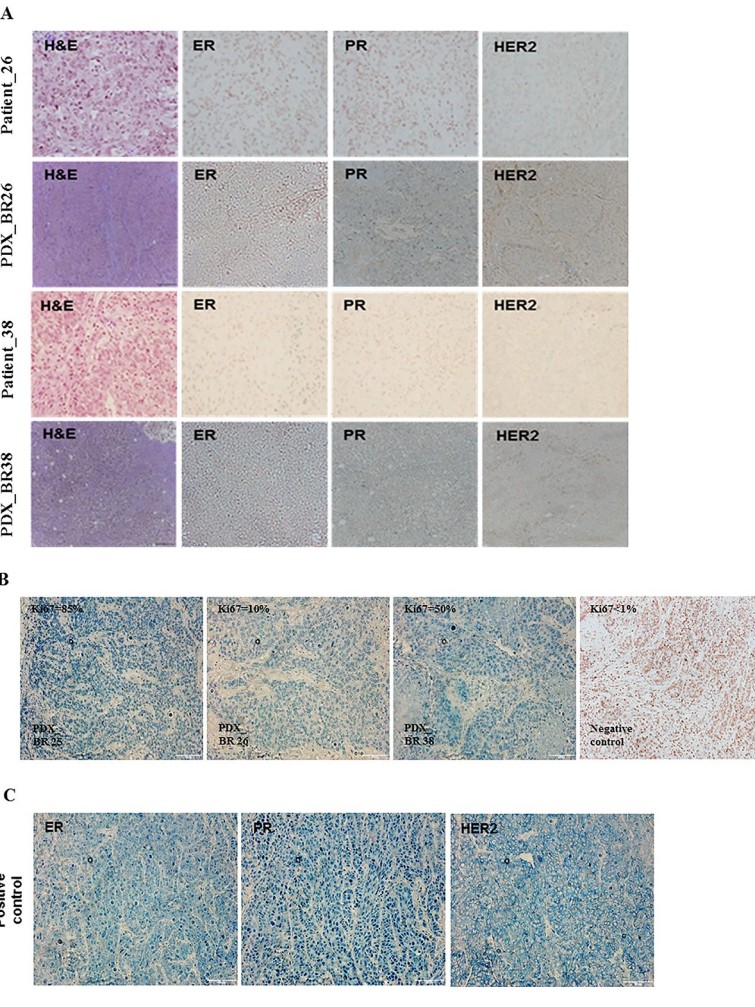

**Fig 2. Histopathological characteristics of successful engraftment of PDX models and patients.** (A) H&E staining and immunohistochemistry of PDX models and primary tumor of patients. (B) Ki67 expression of PDX models. (C) Positive control of ER, PR and HER2 in immunohistochemistry.

Immunohistochemistry revealed Ki67 expression in the PDX models Fig 2B. ER, PR, and HER2 tumor samples were positive controls. (Fig 2C)

The establishment periods of F1-F3 interval are shown in Fig 3A. The median establishment period from F1 to F3 was 221 days (range 93–550 days). The PDX models of residual TNBC with neoadjuvant chemotherapy (Neo TNBC) showed shorter establishment periods than primary TNBC (p = 0.02; Fig 3B). The F2-F3 interval was significantly shorter than the F1-F2 interval (p = 0.00002; Fig 3B). The shortest interval of establishment period was the F2-F3 interval of Neo TNBC models (median 62 days, range 41–158 days; Fig 3B).

In univariate analysis, T-stage, histologic grade, estrogen receptor status, and Ki67 levels were statistically significant (p<0.05) (Table 1). Presence of BRCA mutation, aggressive diseases status, or death were related to successful PDX models (p<0.05). Hormone and HER2 positive cases made up a small portion of total cases (n = 18, 21.7%), and the results were not significant in hormone and HER2 positive cases. All successful PDX models were made from TNBC and Neo TNBC cases.

In 65 TNBC cases, histologic grade, Ki67 level, presence of BRCA mutation, aggressive diseases status, and death were statistically significant on univariate analysis (p<0.05; Table 2).

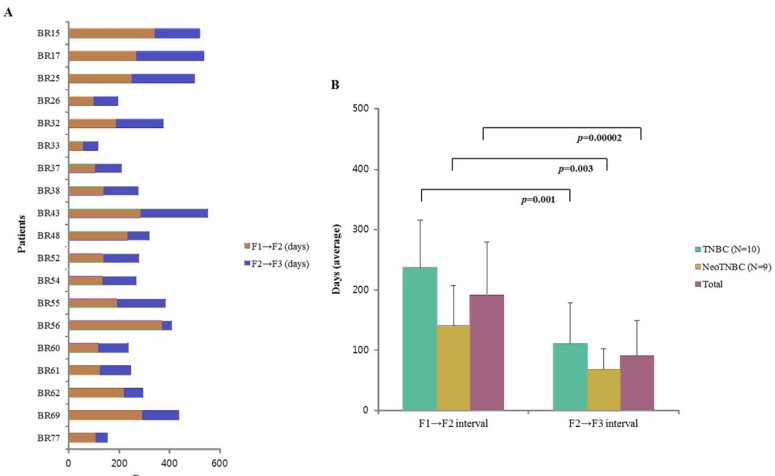

**Fig 3. Successful engraftment of F1 to F3 in PDX models.** (A) Establishment periods of all successful PDX models (B) F1-F2 and F1-F3 intervals according to status of neoadjuvant chemotherapy.

**Table 1. Analysis of factors related to PDX models.**

| Factors | | PDX status | | |
|---|---|---|---|---|
| | | **Failure (%)** | **Success (%)** | ***p*-value** |
| **Age (years)** | <**60** | 43 (67.2%) | 14 (73.7%) | 0.592 |
| | ≥**60** | 21 (32.8%) | 5 (26.3%) | |
| **T stage** | -T1 | 34 (56.7%) | 5 (26.3%) | 0.038 |
| | **T2** | 22 (36.7%) | 10 (52.6%) | |
| | **T3-T4** | 4 (6.7%) | 4 (21.1%) | |
| **Nodal status** | **Negative** | 40 (65.6%) | 8 (47.1%) | 0.165 |
| | **Positive** | 21 (34.4%) | 9 (52.9%) | |
| **Histologic grade** | I/II | 33 (56.9%) | 3 (15.8%) | 0.002 |
| | III | 25 (43.1%) | 16 (84.2%) | |
| **ER** | **Negative** | 51 (79.7%) | 19 (100.0%) | 0.032 |
| | **Positive** | 13 (20.3%) | 0 (0.0%) | |
| **PR** | **Negative** | 55 (85.9%) | 19 (100.0%) | 0.083 |
| | **Positive** | 9 (14.1%) | 0 (0.0%) | |
| **HER2** | **Negative** | 56 (87.5%) | 19 (100.0%) | 0.105 |
| | **Positive** | 8 (12.5%) | 0 (0.0%) | |
| **Ki67 (%, n = 79)** | <**39 (n = 41)** | 38 (62.3%) | 3 (16.7%) | 0.001 |
| | ≥**39 (n = 38)** | 23 (37.7%) | 15 (83.3%) | |
| **BRCA mutation** | **Absent** | 61(96.9%) | 15(78.9%) | 0.005 |
| | **Present** | 2(3.1%) | 4(21.1%) | |
| **Survival status** | Live | 62(96.9%) | 15(78.9%) | 0.008 |
| | Death | 2(3.1%) | 4(21.1%) | |
| **Aggressive diseases*** | **Absent** | 54 (87.5%) | 10 (63.2%) | 0.004 |
| | **Present** | 10 (12.5%) | 9 (36.8%) | |

ER, estrogen receptor; PR, progesterone receptor; HER2, human epidermal growth factor receptor; Ki67: cell proliferation index

* Aggressive diseases were considered to be progressive diseases during neoadjuvant chemotherapy, recurrent, and metastatic disease

**Table 2. Analysis of factors related to TNBC PDX models.**

| Factors | | PDX status | | |
|---|---|---|---|---|
| | | **Failure (%)** | **Success (%)** | ***p*-value** |
| **Age(years)** | <60 | 29 (63.0%) | 14 (73.7%) | 0.410 |
| | ≥60 | 17 (37.0%) | 5 (26.3%) | |
| **T stage** | -T1 | 24 (54.5%) | 5 (26.3%) | 0.070 |
| | T2 | 17 (38.6%) | 10 (52.6%) | |
| | T3-T4 | 3 (6.8%) | 4 (21.1%) | |
| **Nodal status** | Negative | 28 (63.6%) | 8 (47.1%) | 0.238 |
| | Positive | 16 (36.4%) | 9 (52.9%) | |
| **Histologic grade** | I/II | 19 (45.2%) | 3 (15.8%) | 0.027 |
| | III | 23 (54.8%) | 16 (84.2%) | |
| **Ki67 (%, n = 79)** | <39 (n = 41) | 23 (52.3%) | 3 (16.7%) | 0.010 |
| | ≥39 (n = 38) | 21 (47.7%) | 15 (83.3%) | |
| **BRCA mutation** | Absent | 43(93.5%) | 14(73.7%) | 0.027 |
| | Present | 3(6.5%) | 5(26.3%) | |
| **Survival status** | Live | 44(95.7%) | 15(78.9%) | 0.034 |
| | Death | 2(4.3%) | 4(21.1%) | |
| **Aggressive diseases*** | Absent | 37 (80.4%) | 10 (52.6%) | 0.023 |
| | Present | 9 (19.6%) | 9 (47.4%) | |

ER, estrogen receptor; PR, progesterone receptor; HER2, human epidermal growth factor receptor; Ki67: cell proliferation index

* Aggressive diseases were considered to be progressive diseases during neoadjuvant chemotherapy, recurrent, and metastatic disease

Tumor size was not significant in TNBC cases (p>0.05). In TNBC cases with neoadjuvant chemotherapy, histologic grade, presence of BRCA mutation, aggressive diseases status, and death were statistically significant (p<0.05) (Table 3). Information of the chemotherapy regimens is presented in S1 Table.

Multivariate analyses showed poor histologic grade and aggressive diseases status were independently associated with the successful engraftment of PDX models (p<0.05; Table 4), and presence of BRCA mutation and death event were marginally associated with successful engraftment of PDX models (p = 0.05; Table 4).

In the chemosensitivity tests, there were no significant differences in the average tumor volume of the three treatment arms in PDX models with wild type *BRCA1* and the deleterious *BRCA1* mutation. However, nearly complete remission in the carboplatin arm and partial remission in the olaparib arm were attained in the PDX model with the novel L1780P *BRCA1* mutation (Fig 4A–4E).

In WGS, c.5339T>C, L1780P in *BRCA1* was continuously detected in buffy coat, primary tumor, F1, F2, and F3 tumors. However, variant allele frequency (VAF) was increased in late passage. (Normal = 0.53, F0 = 0.63, F1 = 0.9, F2 = 1.0, F3 = 1.0). There was no pathogenic germline mutation in buffy coat, primary tumor and PDX models, other than in L1780P in WGS. In TCGA data, there was no c.5339T>C, L1780P variant among in *BRCA1* mutations. [38] When we investigated the L1780P mutation in a Korean database reported in a previous study, there was no L1780P mutation in the normal population. [32]

## Discussion

We demonstrated that the successful establishment of PDX models from TNBC patients is related to poor clinicopathologic factors including large tumor size, poor histologic grade, and

**Table 3. Analysis of factors related to TNBC PDX models with neoadjuvant chemotherapy.**

| Factors | | PDX status | | |
|---|---|---|---|---|
| | | Failure (%) | Success (%) | *p*-value |
| Age(years) | <60 | 22 (73.3%) | 8 (88.9%) | 0.331 |
| | ≥60 | 8 (26.7%) | 1 (11.1%) | |
| T stage | -T1 | 13 (44.8%) | 2 (22.2%) | 0.066 |
| | T2 | 13 (44.8%) | 3 (33.3%) | |
| | T3-T4 | 3 (10.3%) | 4 (44.4%) | |
| Nodal status | Negative | 16 (55.2%) | 2 (25.0%) | 0.131 |
| | Positive | 13 (44.8%) | 6 (75.0%) | |
| Histologic grade | I/II | 16 (59.3%) | 1 (11.1%) | 0.012 |
| | III | 11 (40.7%) | 8 (88.9%) | |
| Ki67 (%, n = 79) | <39 (n = 41) | 17 (58.6%) | 2 (22.2%) | 0.056 |
| | ≥39 (n = 38) | 12 (41.4%) | 7 (77.8%) | |
| BRCA mutation | Absent | 28(93.3%) | 6(66.7%) | 0.036 |
| | Present | 2(6.7%) | 3(33.3%) | |
| Survival status | Live | 28(93.3%) | 6(66.7%) | 0.036 |
| | Death | 2(6.7%) | 3(33.3%) | |
| Aggressive diseases* | Absent | 23 (76.7%) | 2 (22.2%) | 0.003 |
| | Present | 7 (23.3%) | 7(77.8%) | |

ER, estrogen receptor; PR, progesterone receptor; HER2, human epidermal growth factor receptor; Ki67: cell proliferation index

* Aggressive diseases were considered to be progressive diseases during neoadjuvant chemotherapy, recurrent, and metastatic disease

aggressive diseases status at the time of tissue sampling. Aggressive diseases, including progressive diseases during neoadjuvant chemotherapy, recurrent disease, and metastatic disease at presentation, were associated with the establishment of successful PDX models on multivariate analysis. A previous study reported that TNBC patients with progressive disease during neoadjuvant chemotherapy had higher xenograft take rate than those with stable diseases or partial response after neoadjuvant chemotherapy (6 of 7 patients, 85.7% vs. 5 of 17 patients, 29.4%) [39]. These findings were similar to our findings. Therefore, tumor tissue from patients with TNBC with aggressive features may be the best candidate for establishing PDX models.

Large tumor size and poor histologic grade were significantly related to stable take rates in successful PDX models. A previous study reported that large tumor size and grade were associated with successful engraftment of PDX in bladder cancer [40]. Similar results have also been reported in pancreatic cancer where large tumor size (≥T3) was shown to be related to successful engraftment of PDX [41]. These studies suggested that large tumor size and poor histologic grade are predictors of successful engraftment of PDX, and this is concordant with our results. The analysis of factors related to the successful engraftment of PDX is crucial because considerable time and funds are consumed making successful PDX models. When developing PDX models, the more suitable the patients enrolled are, the less time and funds are required in order to achieve engraftment. The results of this study will be helpful to improve pre-clinical research. Furthermore, genomic research and drug tests in successful PDX models are necessary to improve the performance of PDX models.

In the current study, the mean F1-F2 interval was longer than the mean F2-F3 interval. In concordance with our results, a previous publication also demonstrated that the F1-F2 interval was longer than the F2-F3 interval in successful breast cancer PDX models [39].

In this study, Neo TNBC PDX models had shorter establishment period intervals than TNBC PDX models. These results suggested that shorter establishment period may be due to

**Table 4. Multivariate analysis of factors relating to successful engraftment of PDX models.**

|  | Factors | *p*-value | OR | 95% C.I. |
|---|---|---|---|---|
| **All cases (n = 83)** | Histologic grade (I/II vs. III) | 0.004 | 7.040 | 1.847–26.836 |
|  | BRCA mutation (absent vs. present) | 0.012 | 7.262 | 1.549–34.034 |
|  | Survival status (live vs. Death) | 0.021 | 8.247 | 1.382–49.444 |
|  | Aggressive diseases (absent vs. present) | 0.006 | 4.860 | 1.577–14.974 |
| **TNBC cases (n = 65)** | Histologic grade (I/II vs. III) | 0.034 | 4.406 | 1.114–17.420 |
|  | BRCA mutation (absent vs. present) | 0.039 | 5.119 | 1.083–24.196 |
|  | Survival status (live vs. Death) | 0.053 | 5.867 | 0.974–35.339 |
|  | Aggressive diseases (absent vs. present) | 0.027 | 3.700 | 1.162–11.783 |
| **Neo TNBC cases (n = 39)** | Histologic grade (I/II vs. III) | 0.030 | 11.636 | 1.269–106.719 |
|  | BRCA mutation (absent vs. present) | 0.056 | 7.000 | 0.952–51.448 |
|  | Survival status (live vs. Death) | 0.056 | 7.000 | 0.952-51-448 |
|  | Aggressive diseases (absent vs. present) | 0.007 | 11.500 | 1.930–68.518 |

the aggressive nature of residual tumors after neoadjuvant chemotherapy, compared to primary tumors [42]. Aggressiveness of tumors has been shown to affect the successful establishment of PDX models and the rapid formation of tumor and tumor growth in the immune-deficient mouse [43, 44]. It was concordant with our study. Thus, in terms of establishment periods for establishing PDX models, residual TNBC after neoadjuvant chemotherapy may be more suitable than primary tumor of TNBC.

Chemosensitivity tests using PDX models were introduced by previous studies [45–47]. However, only a few studies reported chemosensitivity tests using PDX models from residual breast cancer after chemotherapy [48]. It has been suggested that BRCA mutations or BRCA-ness are potential targets of PARP inhibitors and platinum agents [33, 49]. Thus, we performed in vivo chemosensitivity tests of PARP inhibitor, olaparib and platinum agent, carboplatin, for PDX models with residual TNBC with/without BRCA1 mutation. A potential benefit from olaparib and carboplatin was evident in the PDX model with the L1780P mutation, but not in the other models. The establishment periods of PDX models remains a major challenge to applying them as in vivo chemosensitivity tests or the baseline study of N-of-1 trials. To overcome this hurdle, appropriate selection criteria for enrollment of patients for the PDX models should be considered. In this study, the median establishment period was 8 months. A limitation of this study is that chemosensitivity tests of the PDX models with L1780P mutation were performed in a single PDX model of one patient. Because the current study was conducted as an initial step of proof-of-concept study of N-of-1 trial, the results provide additional information about the potential application of chemosensitivity tests using PDX models. More chemosensitivity tests of PDX models for a larger number of L1780P mutation cases may strengthen the implications of further investigations of BRCA1 L1780P mutation olaparib and carboplatin drug-sensitivity.

In the current study, histopathological characteristics of tumors in successful PDX models were concordant with primary tumors. Interestingly, the sequencing analyses for the PDX model with L1780P mutation showed the VAF of the mutation was different among passages. This suggests that genetic alteration may occur throughout the passages. However, full genomic analyses were not performed in either primary tumors or successful PDX models—which was a limitation of this study. One of the advantages of PDX models is their ability to reflect patients' genomic and clinicopathologic characteristics. Therefore, sequencing data from patients and PDX models is essential to examine the similarity of the genomic landscape between the two [48, 50, 51].

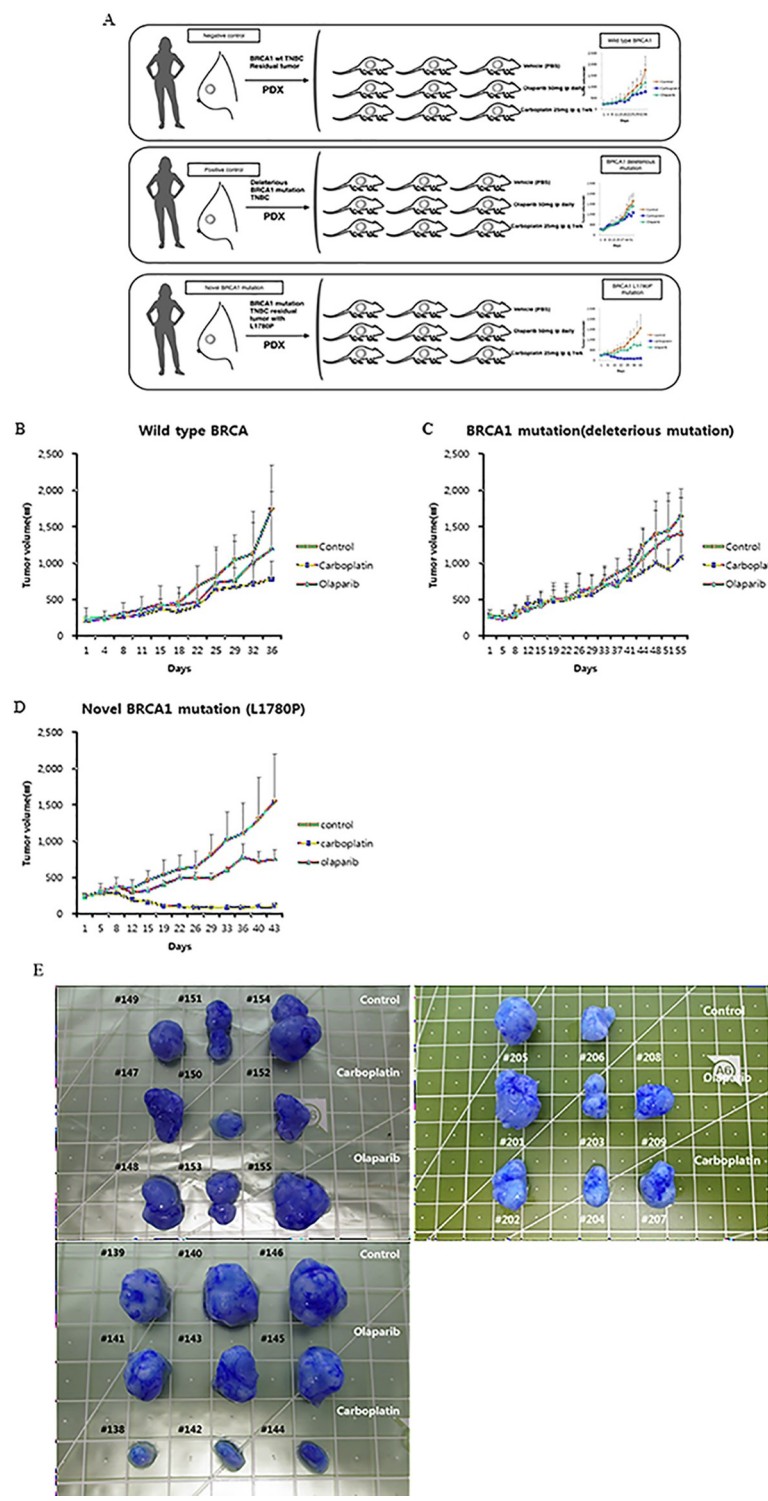

**Fig 4. Scheme and results of chemosensitivity tests using PDX models.** (A) PDX models were derived from patients who had BRCA1 mutation (n = 2) or wild-type BRCA1 (n = 1). Each model comprise nine mice, which were divided into three groups (1 group = 3 mouse, two single treatment groups and one vehicle group). (B-E) When implanted tumor reached an average size of 200–250 mm$^3$ (Volume = 0.5×Length×Width^2), the chemosensitivity test was performed. Olaparib (50mg/kg, once a daily) and carboplatin (25mg/kg, once a weekly) were administered by an intra-peritoneal (i.p.) route. Phosphate buffered saline was used to as a vehicle.

PDX models often utilize immune deficient mouse to avoid rejection of the human tumor graft by the mouse immune system. For this reason, PDX models using immune deficient mouse cannot reflect crosstalk of the immune system and human tumors. A previous study introduced a PDX model using humanized mice that was developed by injection of human hematopoietic cells into the mice to overcome this limitation of PDX models using immune deficient mice. [52, 53]

In further studies, it will be necessary to perform genomic sequencing and analysis of sequencing data in all successful PDX models. Additional experiments are necessary for the two suspected lymphoma to evaluate the mechanism of formation of lymphoma in PDX models [54].

## Conclusions

The current study showed the feasibility of establishing PDX models using residual TNBC after neoadjuvant chemotherapy. Patients who have aggressive diseases status, large tumors, and poor histologic grade are ideal candidates for developing successful PDX models. This study also showed the potential usefulness of in vivo chemosensitivity tests for tumors with targetable biomarkers, particularly tumors with a novel pathologic mutation, L1780P. Further genomic analyses of the PDX models may shed light on tumor progression and drug resistant mechanisms in TNBC.

## Supporting information

**S1 Table. Information on the chemotherapy regimens in the patients for application of PDX establishment.**
(XLSX)

## Acknowledgments

A part of the abstract of the study was presented in the poster session at the Global Academic Programs (Gap) conference, 2018, Stockholm, Sweden.

## Author Contributions

**Formal analysis:** Jee Ye Kim, Seho Park, Joo Heung Kim, Hyun Ju Han, Yeon A. Choi, Ae Ran Choi.

**Writing – original draft:** Hyung Seok Park, Jeong Dong Lee.

**Writing – review & editing:** Joo Hyuk Sohn, Seung Il Kim.

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
