## [Decision Letter · Decision Letter 0]

27 Jun 2019

PONE-D-19-14969

Chemosensitivity test in triple negative breast cancer patient-derived xenograft model

PLOS ONE

Dear Seung Il Kim,

Thank you for submitting your manuscript to PLOS ONE. After careful consideration, we feel that it has merit but does not fully meet PLOS ONE’s publication criteria as it currently stands. Therefore, we invite you to submit a revised version of the manuscript that addresses the points raised during the review process.

We would appreciate receiving your revised manuscript by 31.08.2019. To enhance the reproducibility of your results, we recommend that if applicable you deposit your laboratory protocols in protocols.io, where a protocol can be assigned its own identifier (DOI) such that it can be cited independently in the future. For instructions see: http://journals.plos.org/plosone/s/submission-guidelines#loc-laboratory-protocols

We look forward to receiving your revised manuscript.

Kind regards,

Harriet Wikman, Ph.D.

Academic Editor

PLOS ONE

Journal Requirements:

2. At this time, we request that you  please report additional details in your Methods section regarding animal care, as per our editorial guidelines: 1) Please provide details of animal welfare (e.g., shelter, food, water, environmental enrichment) 2) please describe any steps taken to minimize animal suffering and distress, such as by administering anaesthetics or analgesics, 3) please include the method of sacrifice and 4) Please describe the care received by the animals after tumor implantation, including the frequency of monitoring and the criteria used to assess animal health and well-being. Thank you for your attention to these requests.

Please also provide additional details regarding participant consent. In the ethics statement in the Methods and online submission information, please ensure that you have specified what type of consent you obtained (for instance, written or verbal).

Additional Editor Comments:

Please pay special attention to the presented figures in terms of picture quality and arrangement.

Reviewers' comments:

Reviewer's Responses to Questions

**Comments to the Author**

1. Is the manuscript technically sound, and do the data support the conclusions?

Reviewer #1: Partly

Reviewer #2: Yes

2. Has the statistical analysis been performed appropriately and rigorously? 

Reviewer #1: Yes

Reviewer #2: Yes

3. Have the authors made all data underlying the findings in their manuscript fully available?

Reviewer #1: Yes

Reviewer #2: Yes

4. Is the manuscript presented in an intelligible fashion and written in standard English?

Reviewer #1: Yes

Reviewer #2: Yes

5. Review Comments to the Author

Reviewer #1: This manuscript is a report of the successful development and characterization of a set of TNBC PDXs. The development of these models provide a good resource for other researchers interested in TNBC. There is not very much new information presented within these studies; it is well known that tumor growth rates increase after the first successful passage in vivo, which is the focus of 3 of the figures.

Comments that should be addressed:

1. It would be helpful to state if this PDX cohort are all from Asians, as this could provide a unique resource for other researchers. Along these lines, it would be very informative to know if any of these PDXs are available to other researchers, and if so, this should be stated.

2. A more descriptive title highlighting the interesting aspect of the manuscript is needed.

3. Figure 2A is poorly produced, the text should fit within the graph.

4. The quality of the H&Es and ER/PR/HER2 IHC from the PDXs is very poor, and new pictures should be taken prior to publication. Positive controls for ER/PR/HER2 are always helpful to confirm that the assay is properly working and that the samples highlighted are truly TNBC.

5. If would be helpful to know if the BRCA1 mutation (L1780P) is found in existing patient datasets (TCGA), and if not, this should be stated.

6. The timing of when the carboplatin was administered to the mice (size of tumor at the start of treatment) should be added to the graphs or the text.

Reviewer #2: The manuscript from Park et al. entitled "Chemosensitivity test in triple negative breast cancer patient-derived xenograft model" investigate the factors influencing the establishment of patient derived xenograft models of triple negative breast cancer. The authors successfully generate 19 TNBC PDX and tested carboplatin versus olaparib treatment in 1 wild-type PDX versus 2 with BRCA1 mutation. They also describe a new BRCA1 mutation which sensitize the PDX to carboplatin treatment.

Establishing better in vivo models to acquire strong and valid pre-clinical data is of major importance in Oncology. The work presented here from Park at al. is therefore definitely suitable for publication in PLOS-ONE, however, some major points need to be adjusted before publication.

Major comments:

A few important caveats should be added to the discussion. First, the fact that more aggressive tumors are more likely to generate PDX is quite obvious but one should mention that for this reason, the cohort of PDX used to model TNBC are biased toward more aggressive tumors and do not recapitulated the variety of a human cohort. Moreover, even if primary TNBC have a lower success rate, they are also very interesting and suitable for drug studies and should be consider to design pre-clinical studies, not just the neo TNBC. Second, because of the nature of PDX and the use of immune deficient mice, it is important to remember that most of the tumor micro environment and interaction to the immune system is missing. The authors should therefore improve their discussion.

The authors state that their TNBC PDX are suitable models to test alterative treatment strategies. However, only conventional treatment is presented here. It will have been nice to see new therapies or maybe some combination with irradiation.

Minor comments:

- The authors should pay attention to the manuscript organization. The figure legends should be grouped at the end and precisely describe the figure without including material and methods data. The figures should be organized in one page. Meaning each panel, like 4A 4B should be presented in one page and have a common legend.

- In the first paragraph, please state how may primary and residual TNBC you have in your study, not just the success rate.

- The legend of the figure 1 should go into figure 5. The chemo-sensitivity results are to be reported in fig.5.

- Define what is F1-F3 in the material and method.

- Provide a table that show which PDX were injected subcutaneous and which ones were fat pad. Does that influence the success rate?

- More information should be provided about the how the cells were extracted from the tissue. Was there a brief culture? How many cells were injected subcutaneously or in the fat pad? Was matrigel used? Or did the authors really injected tissue as stated in their material and methods? in that case please provide the size of the tissue and how it was chosen.

- Fig 3. Add a positive control for the staining presented and add Ki67

- Table1 and table 2 have the same title. Clarify which cases were used for each table. If table 2 represent a subset of table 1, it must show in the title.

- Fig5. The graph axis cannot be seen. Please state in the figure legend how many mice were used per group.

- Please provide information on the treatment received by the patients for the neo-PDX. Can the authors see any correlation between the treatment and how fast or successfully the PDX grow?

6. PLOS authors have the option to publish the peer review history of their article (what does this mean?). If published, this will include your full peer review and any attached files.

Reviewer #1: No

Reviewer #2: No

---

## [Author Response · Author response to Decision Letter 0]

22 Oct 2019

Journal Requirements:

2. At this time, we request that you please report additional details in your Methods section regarding animal care, as per our editorial guidelines: 

1) Please provide details of animal welfare (e.g., shelter, food, water, environmental enrichment) 2) please describe any steps taken to minimize animal suffering and distress, such as by administering anaesthetics or analgesics, 3) please include the method of sacrifice and 4) Please describe the care received by the animals after tumor implantation, including the frequency of monitoring and the criteria used to assess animal health and well-being. Thank you for your attention to these requests.

- Thank you for your comment. All requests about animal welfare were described in materials and methods. 

Please also provide additional details regarding participant consent. In the ethics statement in the Methods and online submission information, please ensure that you have specified what type of consent you obtained (for instance, written or verbal).

Additional Editor Comments:

Please pay special attention to the presented figures in terms of picture quality and arrangement.

- Thank you for your comment. We pay attention to picture quality and re-arrangement of figures.

 

Reviewer #1: 

This manuscript is a report of the successful development and characterization of a set of TNBC PDXs. The development of these models provide a good resource for other researchers interested in TNBC. There is not very much new information presented within these studies; it is well known that tumor growth rates increase after the first successful passage in vivo, which is the focus of 3 of the figures.

Comments that should be addressed :

1. It would be helpful to state if this PDX cohort are all from Asians, as this could provide a unique resource for other researchers. Along these lines, it would be very informative to know if any of these PDXs are available to other researchers, and if so, this should be stated.

- A process of our PDX model is now on patenting. When the patent process is completed, we would like to provide our PDX models to other researchers.

2. A more descriptive title highlighting the interesting aspect of the manuscript is needed.

- We revised the title as establishment and chemosensitivity test in triple negative and BRCA mutated breast cancer patient-derived xenograft models.

3. Figure 2A is poorly produced, the text should fit within the graph. 

- Thank you for your comment. We added a new figure and revised the text.

4. The quality of the H&Es and ER/PR/HER2 IHC from the PDXs is very poor, and new pictures should be taken prior to publication. Positive controls for ER/PR/HER2 are always helpful to confirm that the assay is properly working and that the samples highlighted are truly TNBC.

- Positive control of ER, PR and HER2 was added and new pictures of IHC were placed. 

5. If would be helpful to know if the BRCA1 mutation (L1780P) is found in existing patient datasets (TCGA), and if not, this should be stated.

- When we looked at the TCGA data, there was no L1780P mutation in BRCA1. It was described in results. We looked at the L1780P mutation in the Korean database. There was no L1780P mutation in normal population.(1)

6. The timing of when the carboplatin was administered to the mice (size of tumor at the start of treatment) should be added to the graphs or the text.

- When tumor reached average size of 200-250㎣, drug (carboplatin or olaparib) administered to mice. This was described in the legend of figure 4.

Reviewer #2: 

The manuscript from Park et al. entitled "Chemosensitivity test in triple negative breast cancer patient-derived xenograft model" investigate the factors influencing the establishment of patient derived xenograft models of triple negative breast cancer. The authors successfully generate 19 TNBC PDX and tested carboplatin versus olaparib treatment in 1 wild-type PDX versus 2 with BRCA1 mutation. They also describe a new BRCA1 mutation which sensitize the PDX to carboplatin treatment.

Establishing better in vivo models to acquire strong and valid pre-clinical data is of major importance in Oncology. The work presented here from Park at al. is therefore definitely suitable for publication in PLOS-ONE, however, some major points need to be adjusted before publication.

Major comments:

A few important caveats should be added to the discussion. First, the fact that more aggressive tumors are more likely to generate PDX is quite obvious but one should mention that for this reason, the cohort of PDX used to model TNBC are biased toward more aggressive tumors and do not recapitulated the variety of a human cohort. Moreover, even if primary TNBC have a lower success rate, they are also very interesting and suitable for drug studies and should be consider to design pre-clinical studies, not just the neo TNBC. 

- Thank you for your comment. It is necessary to establish the PDX model from hormone and HER2 positive patients. The figure 1 presented that we attempted to establish the PDX model from hormone and HER2 positive patients. Recently, one PDX model was successfully established from ER, PR and HER2 positive breast cancer patient. (Data not shown) To make successful establishment of PDX models from hormone and HER2 positive patients will be continue.

We agreed with your opinion. We are planning to conduct additional pre-clinical study which uses the suitable drug treatment for primary TNBC without neoadjuvant chemotherapy will be performed in the near future.

Second, because of the nature of PDX and the use of immune deficient mice, it is important to remember that most of the tumor micro environment and interaction to the immune system is missing. The authors should therefore improve their discussion. 

- We agreed your opinion. Using the immune deficient mice was major limitation of our PDX models. We described it in discussion.

The authors state that their TNBC PDX are suitable models to test alterative treatment strategies. However, only conventional treatment is presented here. It will have been nice to see new therapies or maybe some combination with irradiation. 

- When we started this pre-clinical study, olaparib and carboplatin were not indicated to tumors with germline BRCA1 mutation. That’s why we only treated monotherapy for our PDX models. As you stated, we agree that the combination treatment with/without irradiation can be a good alternative treatment option for TNBC. These new regimens will be tested in the further study.

Minor comments:

- The authors should pay attention to the manuscript organization. The figure legends should be grouped at the end and precisely describe the figure without including material and methods data. The figures should be organized in one page. Meaning each panel, like 4A 4B should be presented in one page and have a common legend.

- Thank you for your comment. Figure 4A and 4B were presented in one page and described a common legend.

- In the first paragraph, please state how may primary and residual TNBC you have in your study, not just the success rate.

- In TNBC patients, 26 primary tumors and 39 residual tumors were used to establishment of PDX models. This was described in materials and methods.

- The legend of the figure 1 should go into figure 5. The chemo-sensitivity results are to be reported in fig.5.

- The legend of the figure 1 went into figure 5.

- Define what is F1-F3 in the material and method.

- Definition of F1-F3 was described in materials and methods.

- Provide a table that show which PDX were injected subcutaneous and which ones were fat pad. Does that influence the success rate?

- Only one tumor was subcutaneously implanted into mice. This PDX model was failed to establishment. It was too small to compare with PDX models which were implanted into fat pad.

- More information should be provided about the how the cells were extracted from the tissue. Was there a brief culture? How many cells were injected subcutaneously or in the fat pad? Was matrigel used? Or did the authors really injected tissue as stated in their material and methods? in that case please provide the size of the tissue and how it was chosen.

- When patients underwent breast surgery, tumor tissue was removed and immediately transferred to animal laboratory. Cubic tissue was minced and directly implanted into mice without delay. It was described in materials and methods.

- Fig 3. Add a positive control for the staining presented and add Ki67

- Positive control and ki67 were added in the figure 2B and 2C.

- Table1 and table 2 have the same title. Clarify which cases were used for each table. If table 2 represent a subset of table 1, it must show in the title.

- The title of table 1 was revised.

- Fig5. The graph axis cannot be seen. Please state in the figure legend how many mice were used per group.

- Three mouse were used per group. This is described in the legend of figure 4.

- Please provide information on the treatment received by the patients for the neo-PDX. Can the authors see any correlation between the treatment and how fast or successfully the PDX grow?

- Supplementary table 1 presented information on the treatment received by the patients. In successful establishment of PDX models from Neo TNBC patient, 6 of 9 patients received AC+taxane chemotherapy and 2 patients received gemcitabine+taxane, taxane chemotherapy. One patient received AC chemotherapy. We analyzed success rates according to treatment regimens. However, we found no correlated between them. For example, as you see the table 1A and 1B, there was no significant association between treatment group and PDX successful engraftment. 

Table 1. Chi-square analysis of regimen in PDX models from patients who received neoadjuvant chemotherapy.

A. Analysis of AC+taxane based chemotherapy and others in PDX models from patients who received neoadjuvant chemotherapy.

 PDX status 

 Failure (%) Success (%) p-value

 AC+taxane based chemotherapy 22(73.3%) 5 (55.6%) 0.416

 Others 8 (26.7%) 4 (44.4%) 

B. Analysis of AC+taxane based, AC based chemotherapy and others in PDX models from patients who received neoadjuvant chemotherapy.

 PDX status 

 Failure (%) Success (%) p-value

 AC+taxane based chemotherapy 23 (76.7%) 5 (55.6%) 0.222

 AC based chemotherapy 4 (13.3%) 1 (11.1%) 

 Others 3 (10%) 3 (33.3%) 

Reference

1. Park JS, Nam EJ, Park HS, Han JW, Lee JY, Kim J, et al. Identification of a Novel BRCA1 Pathogenic Mutation in Korean Patients Following Reclassification of BRCA1 and BRCA2 Variants According to the ACMG Standards and Guidelines Using Relevant Ethnic Controls. Cancer Res Treat. 2017;49(4):1012-21.

---

## [Editor Report · Decision Letter 1]

29 Oct 2019

Establishment of chemosensitivity tests in triple-negative and BRCA-mutated breast cancer patient-derived xenograft models

PONE-D-19-14969R1

Dear Dr. Kim,

We are pleased to inform you that your manuscript has been judged scientifically suitable for publication and will be formally accepted for publication once it complies with all outstanding technical requirements.

With kind regards,

Harriet Wikman, Ph.D.

Academic Editor

PLOS ONE

Additional Editor Comments (optional):

none

---

## [Editor Report · Acceptance letter]

8 Nov 2019

PONE-D-19-14969R1 

Establishment of chemosensitivity tests in triple-negative and BRCA-mutated breast cancer patient-derived xenograft models 

Dear Dr. Kim:

I am pleased to inform you that your manuscript has been deemed suitable for publication in PLOS ONE. Congratulations! Your manuscript is now with our production department. 

With kind regards,

on behalf of

Dr Harriet Wikman 

Academic Editor

PLOS ONE